# Causal Relationships Between the Use of AI, Therapeutic Alliance, and Job Engagement Among Psychological Service Practitioners

**DOI:** 10.3390/bs15010021

**Published:** 2024-12-29

**Authors:** Boshra A. Arnout, Sami M. Alshehri

**Affiliations:** 1Department of Psychology, College of Education, King Khalid University, P.O. Box 2380, Abha 62521, Saudi Arabia; 2Department of Psychology, College of Arts, Zagazig University, Zagazig 44511, Egypt; 3Department of Learning and Instructor, College of Education, King Khalid University, P.O. Box 8685, Abha 61492, Saudi Arabia; smshehrie@kku.edu.sa

**Keywords:** PLS-SEM, AI, therapeutic alliance, job engagement, mental health service providers

## Abstract

Despite the significant increase in studies on AI applications in many aspects of life, its applications in mental health services still require further studies. This study aimed to test a proposed structural model of the relationships between AI use, therapeutic alliance, and job engagement by PLS-SEM. The descriptive method was applied. The sample consisted of (382) mental health service providers in Saudi Arabia, including 178 men and 204 women between 25 and 50 (36.32 ± 6.43) years old. The Artificial Intelligence Questionnaire, the Therapeutic Alliance Scale, and the Job Engagement Scale were applied in this study. The results showed the structural model’s predictability for using AI and the therapeutic alliance in predicting job engagement and explaining the causal relationships between them compared to the indicator average and linear models. The study also found a strong positive overall statistically significant effect (*p* < 0.05) of the use of AI on therapeutic alliance (0.941) and job engagement (0.930) and a positive overall average statistically significant effect (*p* < 0.05) of the therapeutic alliance on job engagement (0.694). These findings indicated the importance of integrating AI applications and therapeutic alliance skills into training and professional development plans.

## 1. Introduction and Theoretical Background

Structural equation modeling (SEM) is a technique used to examine interactions between constructs and indicators ([24]). The variance-based structure equation model (CB-SEM) and the partial least squares structure equation model (PLS-SEM) are the two approaches that researchers typically employ when estimating structural equation models: PLS-SEM is a predictive causal technique that prioritizes prediction in estimating models to provide causal interpretation, whereas CB-SEM mainly emphasizes theories ([24]; [55]). [26] ([26]) defined PLS-SEM as a structural equation modeling technique that uses partial information instead of applying it to the entire model to explain the common variances of indicators, which helps predict and explain the intended outcomes through in-sample and out-of-sample measures rather than fitting the model.

In recent years, the widespread use of structural equation modeling using PLS has been observed as many researchers have become familiar with this method ([22], [25]; [46]; [51]). Although investigations on the PLS-SEM have gained momentum over the past decade in many disciplines ([4]; [21]; [55]), this method is still unclear to many researchers in psychology ([38]; [45]; [57]).

### 1.1. AI and Mental Health Services

AI is a rapidly advancing and promising field and is believed to achieve significant societal benefits, such as eliminating disease and poverty ([50]; [64]; [66]). Research in AI is concerned with enabling computers to simulate human thinking ([41]). AI can be defined through the concept of intelligent systems introduced by AI. There are many definitions of AI, and most of these definitions can be summarized as follows: (1) technologies that include thinking capabilities similar to those found in humans, (2) technologies that perform actions similar to humans, (3) technologies that can reason, and (4) technologies that can act rationally ([49]; [61]).

The authors of the current study define using AI by mental health professionals as using AI systems that simulate human intelligence to perform tasks in professional job practice and provide specialized services to their clients while maintaining privacy and confidentiality without violating professional work ethics.

AI is frequently used in the clinical and healthcare sectors to support medical decisions, provide new information, improve patient engagement and diagnostic judgments, model behaviors and practices, and enhance the efficacy of therapeutic systems in professional job practice ([7]; [16]; [31]). Also, AI has many applications in counseling and therapy, most often in assessment tasks ([7]; [16]; [31]; [32]; [49]; [61]). It has also been used in client monitoring where intelligent systems are applied to assist workers by identifying known factors that can predict the outcomes of the therapeutic process and detect clients’ moods ([48]; [62]). AI applications can play an essential role in therapy by providing insights into the client-therapist interaction and the client’s psychological states ([5]).

The effects of AI on psychiatrists’ duties, particularly concerning prescriptions, have also been demonstrated ([61]). One issue with applying AI to mental health is that although AI is being used in healthcare, various stakeholder groups (including government policymakers, hospital administrators, therapists, and managers of IT companies) occasionally express conflicting opinions about this adoption of AI in practice. AI can assist mental health workers through hybrid intelligence systems, which combine computer and human intelligence ([6]). On the other hand, there are possible drawbacks, including lost jobs and privacy concerns ([5]). In the same context, AI has a favorable and positive impact on the healthcare sector, particularly in hospitals, regarding accuracy, efficiency, and quality of care. However, AI may potentially have unfavorable effects on privacy, ethics, and the dehumanization of healthcare ([34]).

Similarly, [53] ([53]) pointed out that professionals who followed a cognitive behavioral method were more effective than others in using AI applications. In particular, cognitive psychotherapists showed a higher belief in the usefulness and desire to use AI in professional practice. In addition, they were more comfortable using AI applications than others.

Conversational AI has a vital role in psychotherapy, including support for individuals with mental health problems who cannot reach care ([54]). Approximately 55% of clients preferred AI-based psychotherapy when it came to their attitudes toward its use in psychotherapy; when asked if they trusted personal data protection, most clients trusted human psychotherapists more than AI-based technologies. Nonetheless, three significant advantages of AI-based therapy were noted: the capacity to discuss embarrassing events, availability at any time, and the ability to communicate remotely. The belief that AI-based therapy systems can enhance themselves based on prior therapeutic outcomes was one of the factors influencing preference for AI-based psychotherapy. Preference for AI-based psychotherapy was also linked to gender, jobs in psychology and technology/engineering; findings imply that greater trust in AI tools and knowledge of the advantages and efficacy of psychotherapy may enhance preference for AI-based psychotherapy ([2]).

AI in counseling is one field that has benefited from the development of computer technology and the Internet. By using AI, counselors face the challenge of transforming computer technology and the Internet into a profitable way to access dynamic, innovative, and future counseling services and the challenge of how this can affect the quality of services. There are four areas where AI can be applied in psychological counseling: technology-based needs assessment, technology-based counseling process, technology-based counseling method development, and technology-based counseling applications’ ethical challenges ([54]).

Recently, [30] ([30]) focused on chatbots that employ cognitive behavioral therapy to investigate psychotherapists’ attitudes, experiences, and perspectives on using AI-based therapeutic chatbots in treatment therapy. Seven participants with backgrounds in cognitive behavioral therapy and mental healthcare participated in deep interviews using a qualitative study approach. Reflective thematic analysis was employed to examine the data utilizing theoretical frameworks such as the universal ethics of social work principles, fidelity theory, and the diffusion of innovation theory. According to the findings, participants worried about ethical dilemmas, the therapeutic alliance, and AI applications that can handle complex emotional requirements; they also acknowledged the advantages of AI-based therapeutic chatbots, such as improved accessibility and consistent therapy.

### 1.2. Artificial Intelligence and Therapeutic Alliance

The most investigated factor in counseling and psychotherapy studies is the therapeutic alliance, which stands for the quality of the therapist–client connection, which, in turn, is connected to the success of therapeutic outcomes and therapy efficiency ([35]). In therapy of adults ([17]), children and adolescents ([33]), and couples and families ([18]), the therapeutic alliance is also a powerful predictor of therapy outcomes. Numerous studies on the role of the alliance in adult individual therapy have demonstrated that it uniquely contributes to the outcome beyond accepting client characteristics and other process variables, like homework compliance and evaluations of competence and commitment ([17]) Similarly, the therapeutic alliance is a professional connection between clients and therapists based on mutual trust and respect, collaboration in therapy duties, and agreement on therapy goals ([65]).

According to the authors of the current study, a therapeutic alliance is an arrangement between a client and a mental health provider that helps build a solid relationship through cooperation, trust, self-awareness, conflict handling, flexible empathy, confidentiality, transparency, and a shared vision and values. These factors also result in positive therapeutic outcomes and client compliance with homework.

Trainee therapists or beginner psychologists encounter several inherent challenges when involving their clients in building therapeutic processes and therapeutic alliances. These challenges include (1) challenges with “personal material”; (2) challenges with certainty, control, and ideal intentions; (3) frustrations with the client’s presentation; (4) challenges with being the center of attention; (5) reactions to perceived exclusion; (6) anxiety about difference; and (7) personal strategies for handling strong, intense emotions ([2]).

Additionally, AI applications are used in both the therapeutic alliance and the empathy of psychotherapists. Just as psychotherapists collect data from various client indicators (e.g., body language, words, eye contact, or case history), AI applications in the psychotherapy process can differentiate how to benefit from various aspects of therapeutic interactions or physiological outcomes to better understand the therapy procedures and the alliance between the therapist and the patient ([12]; [44]).

Likewise, the results of [9]’s ([9]) study showed how AI applications contribute to developing a therapeutic alliance with a conversational agent, how this alliance evolves, and how it takes shape. AI can define weekly and overall goals, aims, and activities for each session to provide adequate mental health interventions; it can help create a more potent therapeutic experience by strengthening the therapeutic alliance’s goal and task sub-point alliance. According to [17] ([17]), AI can only be useful in a great therapist’s toolbox at its best, and generating paperwork for something as basic as setting up appointments might lessen and decrease the workload for the therapist.

### 1.3. AI and Job Engagement

AI technology in the workplace changes job design, one aspect of job engagement that impacts work quality. Technology might also be detrimental since it could replace people who carry out work-related duties ([43]). In various industries, including education, AI technologies can forecast employee engagement ([37]).

[37] ([37]) defined job engagement as the degree to which people succeed at their job, have the drive and interest to succeed there, feel psychologically at ease carrying out their job responsibilities, are willing to put in extra effort without getting paid to complete the tasks and work that are assigned to them, and try to improve themselves and their skills.

According to the current study’s authors, job engagement for mental health service providers is defined as a positive mental state associated with the profession of providing mental health services to clients; it includes positive feelings of vitality, resilience, perseverance, and energy when working with clients and completing the tasks assigned to them in professional practice, which may lead to a feeling of importance, enthusiasm, inspiration, pride, dedication, overcoming obstacles, complete concentration, deep absorption, and well-being at work.

Utilizing AI technologies improves workers’ mental health, boosts their energy in the workplace, and promotes their sense of competence ([67]). AI can affect employee productivity in several ways ([1]; [13]; [15]; [29]; [39]; [60]; [63]). The resource-based theory has offered a theoretical framework for comprehending how companies can invest in employee skill development, implement efficient technology systems using AI applications, and enhance HR management procedures to use their resources to gain a sustained competitive advantage. Institutions can establish a work environment that encourages employee engagement and boosts output and performance ([8]).

From the above, there is a clear interest in investigating artificial intelligence applications and their role in providing health services. However, their role in developing the therapeutic alliance and job engagement for mental health service providers is still being examined as researchers in the Arab environment lack interest in studying this object.

## 2. The Study Hypothesis

Theoretical literature indicates that health sector workers cope with high stress and burnout. [3]’s ([3]) study found high job burnout among health sector employees and students. Also, [18] ([18]) showed that counselors have a high level of psychological stress that decreases their productivity ([42]). These findings indicate the need to find ways to relieve their increasing work burden ([5], [6]). Many research findings also demonstrate that using AI applications to complete activities at work improves psychological well-being, boosts productivity, decreases workload, and promotes sustainable growth and efficiency ([6]; [15]; [29]; [39]; [63]). Additionally, the Sultan Qaboos University Student Counseling Center’s third conference recommended improving the effectiveness of specialists, psychotherapists, and educational counselors using contemporary technology to help them overcome the difficulties associated with counseling. This aligns with the objectives of the “Vital Society” axis within the Kingdom’s National Vision 2023 ([6]). Accordingly, the authors of the current study assumed that the use of AI is related to job engagement among mental health service providers.

Additionally, many studies ([1]; [7]; [13]; [60]; [16]; [31]) examined the use of AI in psychological services and the divergent opinions of professionals in this field regarding the advantages and disadvantages of utilizing AI systems to carry out their duties.

Similarly, previous studies showed that AI could be used to enhance the quality of services ([34]) and task completion ([59]), support diagnostic decisions, practices, and behavior modeling, and improve the efficacy of therapy ([7]; [16]; [31]). AI can also be used for most tasks, such as assessment, diagnosis, task improvement, and therapy effectiveness ([27]). Additionally, findings from many previous studies ([33]; [18]; [35]) indicate that the therapeutic alliance is a powerful predictor of therapy outcomes and that building a therapeutic alliance between mental health service providers and clients is challenging ([11]), and AI can help in coping with these challenges ([17]; [12]; [9]). According to these results, the current study’s researchers assumed that there was a correlation between the use of AI and therapeutic alliance among mental health service providers.

It is clear from the findings of previous studies on AI and its role in providing mental health services and the challenges facing psychological services providers in developing the therapeutic alliance, as well as the results of studies on the role of AI in employee engagement, that there is a gap in examining the inter-relationships between the use of artificial intelligence, therapeutic alliance, and job engagement among mental health service providers.

Based on the above, the current study’s authors developed the suggested theoretical model in Figure 1 to test the following hypothesis: “Use of AI, therapeutic alliance, and job engagement form a structural model that explains the causal relationships between them among mental health service providers”.

According to this suggested model, depicted in Figure 1, the use of AI is thought to impact job engagement and therapeutic alliance directly. The therapeutic alliance also directly impacts the study sample members’ participation in job engagement, and by mediating the therapeutic alliance variable, the use of AI indirectly impacts job engagement. The partial least squares approach will be used to construct structural equations to verify and test this suggested hypothesized structural model.

## 3. Research Methodology and Procedures

### 3.1. Methodology and Participants

The proposed model, which included the use of AI, therapeutic alliance, and job engagement, was tested using the descriptive approach. Structural equations were applied using the partial least squares method. The study population consisted of psychiatrists, psychologists, and counselors. The criteria for selecting participants in the current study included working as a mental health service provider in the health sector, having at least two years of work experience, and still working as a mental health provider.

An online link, including the study tools and sociodemographic data questionnaire, was sent to 621 people randomly selected from the study population, and 409 of them replied. After checking their responses, 27 respondents were excluded because they did not meet the selection criteria; thus, the final study sample consisted of (382) mental health providers (178 men and 204 women), including psychiatrists (56), psychologists (139), and counselors (187) (Figure 2), with ages ranging from 25 to 50 (36.32 ± 6.43), 71 of them having 2 to 5 years of work experience as mental health providers, 147 having 6 to 9 years of work experience, and 164 of them having more than 10 years of work experience. All study participants indicated their written consent to participate by checking the box on the question: Do you wish to participate in the current study? This was after the institution had approved the study.

### 3.2. Measures

#### 3.2.1. Use of Artificial Intelligence Questionnaire (UAIQ)

The current study’s authors developed a scale with twelve items divided across four dimensions after reviewing theoretical frameworks and earlier research on artificial intelligence. Each dimension had three items answered on a five-point Likert scale from strongly agree to disagree strongly. Using AI in diagnosis and evaluation was the first dimension, which measured how mental health service providers use AI tools to diagnose and assess patients before creating a suitable treatment plan. The second dimension evaluated mental health service providers’ usage of AI applications to support clients, either fully autonomously such that AI systems are the only source of support or incorporated into some sessions with case follow-up. Furthermore, evaluating the use of AI systems by mental health service providers by applying the modeling technique to the desired practices and behaviors that the client hoped to learn and acquire during intervention sessions was the third dimension. The fourth dimension measured the use of AI to assist mental health service professionals in planning and directing therapeutic sessions, including scheduling sessions and managing time.

#### 3.2.2. Therapeutic Alliance Scale (TAS)

Utilizing these theoretical frameworks and previous studies on the therapeutic alliance, the current study’s authors developed a 15-item therapeutic alliance measure. A five-point Likert scale, ranging from strongly agree to disagree strongly, was used to answer all positively worded questions. Each of the five dimensions had an equal distribution of the scale components. As the first component of TAS, “trust and cooperation” assessed how well mental health service providers build relationships with their clients based on mutual respect, trust, acceptance, and cooperation to assist them in reaching their objectives and resolving their issues. The second dimension was “Self-awareness and flexible empathy”, which evaluated how well mental health service providers could use their profound influence and skillful communication to help clients identify their strengths and weaknesses through dialogue and give them strategies to deal with their fears or worries. The third was “shared values”, which assessed how well clients and mental health service providers agreed on therapy goals, potentials and capabilities, confidentiality and openness, and roles and obligations to guarantee effective communication and guide efforts and interventions to reach the objectives. The fourth dimension, “client compliance”, evaluated how well clients adhered to instructions, finished their homework, participated fully in sessions, were adaptable and receptive to service providers, and kept to their appointment schedules. Nonetheless, “conflict management” was the fifth component, which assessed the capacity of mental health service providers to address pressing issues with clients, settle disputes amicably and adaptably, and turn disagreements into chances to improve their clients’ relationships.

#### 3.2.3. Job Engagement Scale (JES)

The current study’s authors developed the job engagement scale for mental health service providers by reviewing theoretical frameworks on job engagement based on the reviewer. The twelve items in JES required responses on a five-point Likert scale, which went from “fully applicable” to “not applicable at all”: (1) energy and vitality, which assessed resilience, energy, and perseverance in carrying out tasks that improve the effectiveness of the services rendered to clients; (2) enthusiasm and dedication, which assessed enthusiasm, inspiration, sense of importance, pride, and challenges in professional practiceThe absorption and persistence (3) measured the mental health service providers’ complete focus on their professional practice and deep immersion, which made them not want to separate or stop working as time passed. Also, the well-being and belonging (4) measured the mental health service providers’ feelings of well-being, happiness, and satisfaction with work, which was positively reflected in their belonging to their job.

### 3.3. Data Analysis

Smart PLS.4.1.0.6 ([47]) applies PLS-SEM to verify the quality of the outer and inner models of the proposed structural models. JASP 0.18.3.0 was used to test the psychometric properties of the study tools by calculating Pearson’s correlation coefficient, omega coefficient, and Cronbach’s alpha coefficient. Mediation analysis was conducted using the bootstrapping test method to ascertain the indirect and total effects. The properties of the proposed structural model and its predictive ability were judged in light of the criteria set by [23] ([23]).

To assess the size of the effect strength, the Cohen criterion (1988) was used: effect sizes were small when they were less than 0.20; they were small when they were between 0.20 and 0.50; they were average when they were between 0.50 and 0.80; they were large when they were between 0.80 and (1.10); they were enormous when they were between (1.10) and (1.50); and they were huge when they were 1.50 or greater ([10]).

## 4. Results

### 4.1. Measuring Validity and Reliability

#### 4.1.1. Using AI Questionnaire Validity and Reliability

Using an AI Questionnaire (UAIQ), internal consistency was tested by calculating Pearson correlation coefficients* between the scores of each item of the scale and each other and the scale as a whole. Table 1 shows the results.

The findings shown in Table 1 indicate that the Pearson correlation coefficients between items were positive and statistically significant (*p* < 0.001) and ranged between 0.636 and 0.920. On the other hand, the Pearson correlation coefficient values between the items and the UAIQ total score varied, were positive and statistically significant, and ranged from 0.777 to 0.943; these results indicated the UAIQ’s internal consistency. The Pearson correlation coefficients were computed using values between items, and their dimension’s overall total scores were also computed. The results of the outcomes are shown in Table 2.

It is clear from Table 2. that all Pearson correlation coefficients between the items of UAIQ and their dimensions were positive and statistically significant (*p* < 0.001), and these values ranged between 0.913 and 0.947

Pearson correlation coefficients were also calculated between the dimensions and the UAIQ total score, and the results are shown in Table 3.

The results in Table 3 demonstrate that all Pearson correlation coefficients were positive and statistically significant (*p* < 0.001). The values of the correlation coefficients between the dimensions and each other ranged between 0.910 and 0.937, while the values of the Pearson correlation coefficients of these dimensions with UAIQ as a whole ranged (0.911–0.932); these results revealed the internal consistency of UAIQ.

Also, Cronbach’s α and McDonald’s ω coefficients were computed to confirm the reliability of UAIQ. As the Cronbach’s α value was (0.940), the results showed that it was equivalent to the McDonald’s ω coefficient. According to these findings, UAIQ has good psychometric properties.

#### 4.1.2. Therapeutic Alliance Scale Validity and Reliability

The internal consistency of the therapeutic alliance scale (TAS) was tested by calculating Pearson correlation coefficients* between the scores of each item of TAS with each other and TAS as a whole. Table 4 shows the results.

Table 4 shows that all Pearson correlation coefficients between the items and the TAS total were positive and statistically significant (*p* < 0.001), and their values ranged from 0.735 to 0.864. On the other hand, the correlation coefficient values between the items and the TAS overall total score ranged from 0.880 to 0.930. Additionally, the Pearson correlation coefficients between items and their dimensions were computed using their dimension’s overall score, and the outcomes results are displayed in Table 5.

It is clear from Table 5 that all Pearson correlation coefficients between the items of TAS were positive and statistically significant (*p* < 0.001), and these values ranged between (0.899 and 0.942).

Also, Pearson correlation coefficients were calculated between the dimensions, and the total TAS score is shown in Table 6.

The results in Table 6 indicate that all Pearson correlation coefficients, whether between the dimensions collectively or within TAS overall scores, were positive and statistically significant (*p* < 0.001). The correlation coefficient values between dimensions ranged from 0.894 to 0.934, and the correlation coefficients between these dimensions and overall TAS ranged from 0.931 to 0.934. These findings demonstrate the internal consistency of TA.

Cronbach’s α and McDonald’s ω coefficients were computed to test the TAS reliability; the results showed that Cronbach’s α value was equal to the McDonald’s ω coefficient, which was (0.928). According to these findings, TAS has strong dependability indices.

#### 4.1.3. Job Engagement Scale

The internal consistency of the Job Engagement Scale (JES) was tested by calculating Pearson correlation coefficients* between the scores of each item and the scale as a whole. Table 7 shows the results.

The findings shown in Table 7 illustrate that the Pearson correlation coefficients between the items and the JES total score were positive and statistically significant (*p* < 0.001), with values ranging from 0.761 to 0.887. On the other hand, the correlation coefficient values between the items and the JES total score ranged from 0.906 to 0.945. Additionally, the Pearson correlation coefficients between items and their dimensions were computed. Table 8 shows the results.

It is clear from Table 8 that all Pearson correlation coefficients between the items of JES were positive and statistically significant (*p* < 0.001), and these values ranged between (0.905 and 0.943).

Pearson correlation coefficients were also calculated between the dimensions and JES total score, and the results are shown in Table 9.

The results in Table 9 indicated that all Pearson correlation coefficients were positive and statistically significant (*p* < 0.001). JES’s internal consistency was demonstrated by the correlation coefficient values between the dimensions, which varied from 0.910 to 0.936, and between the dimensions and JES total scores, which ranged from 0.917 to 0.939. Also, Cronbach’s α and McDonald’s ω coefficients were computed to confirm the validity and reliability of JES. The findings showed that McDonald’s ω was (0.931) and Cronbach’s α was (0.930). According to these findings, JES’s substantial dependability metrics have good psychometric properties.

### 4.2. Testing the Causal Relationships of the Use of AI, Therapeutic Alliance, and Job Engagement

To verify the validity of the hypothesis, which states that “the use of AI, therapeutic alliance, and job engagement among mental health service providers constitute a structural model to explain the causal linkages between them”, and test the proposed model’s properties (see Figure 1), the therapeutic alliance variable’s indirect effects and mediating role in the connection between the use of AI and job engagement were examined by structural equation modeling utilizing the partial least squares (PLS-SEM) and bootstrap approaches. The [23] ([23]) technique yielded the following results:

#### 4.2.1. First: Evaluating the Measurement Reflective Model

The evaluation of the measurement reflective model was performed in four steps as follows:

***Step one: Verifying the loading of AI, therapeutic alliance, and job engagement*** indicators. To provide an item with adequate reliability, [23] ([23]) suggested loading more than (0.708), which shows that the construct explains more than 50% of the indicator’s variance. Figure 1 shows the indicators’ loadings.

All of the study variables’ indicators met the ideal criterion established by [23] ([23]) who stated that the loadings of the indicators should be greater than (0.708), as seen from the data displayed in Figure 3; the findings indicate that every indicator of the outer model’s three structures was higher than this threshold; the use of AI indicators ranged (0.774–0.944); therapeutic alliance indicators ranged (0.877–0.931); and job engagement indicators ranged between 0.906 and 0.945. These results indicate no bias in the study participants’ responses.

***Step two: Testing the internal consistency reliability:*** According to the results, the composite reliability values for job engagement, therapeutic relationship, and use of AI were 0.834, 0.872, and 0.921, respectively. Since none of them achieved a value of (0.95) or higher, they were all within the excellent range established by [23] ([23]), confirming the reflective model’s internal consistency.

***Step three: Verifying convergent validity:*** The results demonstrate that the average variance extracted (AVE) for using AI, therapeutic alliance, and job engagement was higher than (0.50), and thus, these values are acceptable and meet the third condition of the structural model according to the methodology of [25] ([25]).

***Step four: Discriminant validity:*** The heterotrait-monotrait ratio (HTMT) was computed to assess discriminant validity. The outcomes are displayed in Table 10.

The findings in Table 10 indicate that the inner model’s heterotrait-monotrait ratio (HTMT) values fell within the acceptable standard range as per the [26] ([26]) criterion, with values ranging from 0.526 to 0.695 and falling short of 0.90. These findings indicate that using AI, therapeutic alliance, and job engagement is free from interference as the relationship between them did not exceed (0.80), which indicates their discriminant validity.

These results of testing the properties of the outer reflective model explained in the four steps above show the model’s validity and reliability, with reliable values within the standard acceptance criteria set by [23] ([23]) for acceptance.

#### 4.2.2. Second: Evaluation of the Inner Structural Model of the Variables

The following procedures were performed to assess the inner structural model of the study variables using the structural equation model and the partial least squares method carried out using the Smart PLS program:

***Step one: Test the variance inflation factor (VIF) through collinearity statistics***. According to the statistics findings, all of the linear inter-relationship (VIF) values for the outer model fall within the acceptable range (the value is greater than or equal to 3 and less than or equal to 5); none of the values exceed 5. The results are shown in Table 11.

Also, the variance inflation coefficients for the inner model were tested, and the results are shown in Table 12.

None of the variance inflation coefficients (VIF) for the inner model exceeded 5, and the findings displayed in Table 12 demonstrate that they were within the optimal range under the [23] ([23]) criterion.

The results of convergent validity and collinearity (VIF) of the use of AI, therapeutic alliance, and job engagement indicate that the proposed model in our study is free of response bias and common method bias.

***Step two: Testing the statistical significance of the paths.*** The results show that all direct, indirect, and total paths between the use of AI, therapeutic alliance, and job engagement were statistically significant (*p* < 0.001). The results are shown in Table 13.

The results presented in Table 13 show that the use of AI in job engagement had a weak direct positive effect (value of 0.277), the use of AI in the therapeutic alliance had a strong direct positive effect (value of 0.941), and the use of AI in job engagement was statistically significant (*p* < 0.05). Additionally, the therapeutic alliance has a positive direct effect on job engagement with a value of (0.694), which has a medium statistical significance (*p* < 0.05).

Similarly, results in Table 13 show that the application of AI in job engagement through the therapeutic alliance had an indirect effect that was positive, medium, and statistically significant (*p* < 0.05) with a value of 0.653. Additionally, these findings demonstrate the therapeutic alliance’s mediating role in the relationship between the use of AI and mental health service providers’ job engagement. Because of this complementary role, the effect of the use of AI was found to be in the same direction (positive), with the value of the impact rising from (0.277) in the direct effect to (0.653).

In addition, the results shown in Table 13 indicate that the therapeutic alliance has an integrative mediating role in the relationship between the use of AI and job engagement among mental health service providers. The direct effect of the use of AI in job engagement was a weak positive effect with a value of 0.277; it had a medium positive indirect effect with a value of (0.653), and then it became a strong positive overall effect with a value of (0.930).


**
*Step three: Evaluating the predictive significance of the structural model.*
**


Coefficient of determination (R-square): The findings showed that the coefficient determination (R^2^) or the ability of the independent variables together (use of AI and the therapeutic alliance) was high in explaining the change in the dependent variable (job engagement) as the value of R^2^ was (0.920), meaning that 92% of the change in the job engagement of mental health service providers can be explained by the use of AI and the therapeutic alliance together. In addition, the effect of the use of AI variable was high in explaining the change that occurred in the therapeutic alliance with a value of (0.885), or that the use of AI can explain 88.50% of the change in the therapeutic alliance. These findings indicate that interventions based on improving the use of AI skills in the professional practice of mental health service providers contribute to developing their ability to establish a therapeutic alliance with clients and increase their job engagement in terms of energy, vitality, dedication, pride in the profession, belonging, and their psychological well-being in work settings.

According to the test’s findings of the strength of the effect (F-square), using AI as an independent variable in the therapeutic alliance as a dependent variable had a very strong statistically significant effect (F^2^), with a value of 7.693. The therapeutic alliance variable had a medium impact on the mental health providers’ job engagement (0.690). The use of AI, however, had a negligible impact on employee engagement (0.110).

Likewise, the blindfolding analysis was used to determine the predictive value Q^2^, which determined the predictive significance. The findings indicated that the root mean square error index (RMSE) values for the job engagement and therapeutic alliance variables were 0.347 and 0.377, respectively, and the Q^2^ values for these variables were 0.882 and 0.861. These findings showed that the independent (use of AI) and dependent (job engagement) variables had considerable predictive relevance. The independent variables also showed predictive importance for the dependent variables (therapeutic alliance and job engagement).

In addition, [57] ([57]) indicated that the predictive value of the model can be evaluated by conducting a cross-validated predictive ability test (CVPAT)/PLS prediction algorithm, in which the predictive ability of the proposed structural model PLS-SEM is compared to the linear model (LM) and against the indicator average index (IA). The results of the CVPAT test, which was conducted to compare the structural model PLS-SEM vs. LM and IA, are shown in Table 14 and Table 15.

The findings in Table 14 demonstrate that the PLS-SEM structural model’s predictive value for the study variables is superior since its average loss is lower than the indicator average. The structural model for the study variables has a significant predictive value (*p* < 0.05) for predicting both the therapeutic alliance and job engagement from the use of AI, as well as predicting job engagement from the use of AI and therapeutic alliance. This is because all of the average loss difference values were negative.

The findings in Table 15 demonstrate that the structural equation model PLS-SEM has a higher predictive value (*p* < 0.05) for predicting both the therapeutic alliance through the use of AI and the job engagement through the use of both AI and the therapeutic alliance than the linear model (LM). This is because the average loss difference was negative, indicating that the structural model predicted the study variables more accurately than the linear model. The results of the structural model validity test also showed that the value of the standardized root-mean-square residuals SRMR was (0.033), the value of the model fit coefficient NIF reached 0.962, and the chi-square value was (1162.182).

## 5. Discussion

The current study aimed to evaluate the hypothesized structural model for the causal links among relationships between job engagement, therapeutic alliance, and the use of AI by employing PLS-SEM. Results showed the reliability of the measurement reflective model as none of the loadings reached a value of (0.95) or higher and achieved the convergent validity criteria as the results showed that all values above (0.50) were acceptable. Also, the structural equation modeling using the partial least squares method demonstrated that the outer model met all measurement criteria as all loadings were higher than (0.708) according to the [23] ([23]) criterion.

Additionally, discriminant validity demonstrated that the study variables were free from interference because all values fell within the permitted range, and none attained a value of 0.90. Also, the results of testing the variance inflation coefficients to evaluate the internal structural model of the variables indicated that the statistics of the inter-relationships, whether for the internal model represented by the latent variables or the external model of the scale items, all fell within the acceptable range (greater than or equal to 3 and less than or equal to 5). Additionally, the use of AI had a weak, positive, direct, and statistically significant effect on job engagement, according to the results of the path coefficients for the direct, indirect, and total effects.

The results also demonstrate that the therapeutic alliance has an indirect effect or a mediating role in the relationship between the use of AI and job engagement of the psychological mental health service providers; these findings align with [12] ([12]) found that AI helps overcome obstacles to develop a therapeutic relationship, which is a reliable indicator of therapy success ([18]; [33]; [35]). Using AI applications in professional practice enhances the feeling of competence and vitality, improves the psychological state ([67]), and increases productivity and motivation for sustainable professional growth ([15]; [29]; [39]; [63]). Also, [20] ([20]) pointed out that a successful therapeutic alliance refers to agreeing with the client on the goals and tasks of therapy, maintaining empathic understanding and involvement with the client, and ensuring that the client effectively performs assigned tasks ([56]). Previous study investigations revealed the ability of mental health apps to support therapeutic alliance ([36]).

Additionally, the alliance between the client and the mental health provider is an active therapeutic component independent of any other therapeutic technique. A shared sense of cooperation, trust, and hope makes the clients expect that the psychotherapist will understand them, tolerate them, and help them understand themselves and feel better. Feeling understood, cared for, and cared for is a prerequisite for revealing oneself openly to a stranger ([56]; [36]).

So, when the alliance is successful, mental health providers can accomplish an amazing and impressive amount of work, which makes them feel comfortable and happy when working with clients ([36]; [28]). Research in mental health has shown a correlation between therapeutic alliances and therapy outcomes ([14]; [19]; [52]; [58]); therefore, we must apply motivational techniques to generate and maintain strong therapeutic alliances, such as AI applications, to enhance the positive results of therapy ([56]).

These results also support the findings of [34] ([34]) who found that AI facilitates completing tasks by mental health professionals. AI is used to help mental health service providers overcome the obstacles they face in gaining the trust and cooperation of clients and complying with duties and instructions ([5]). Previous studies have shown that AI is used in most tasks related to providing mental health services, such as assessment and diagnosis ([49]), supporting diagnostic decisions, improving the effectiveness of treatment systems, and modeling practices and behaviors ([7]; [16]; [40]; [61]).

Based on these findings, we conclude that using AI applications in professional practice improves clients’ mental health service providers’ commitment to responsibilities, duties, and instructions; increases their confidence and cooperation with mental health service providers’ clients; improves their ability to handle potential conflicts; and, also, increases awareness of their own needs and goals, which decreases time and effort and, consequently, improves their well-being, enthusiasm, and dedication to work, the client’s satisfaction with them, and the effectiveness of treatment therapy outcomes ([5]; [7]; [16]; [19]).

Furthermore, the current results demonstrate a statistically significant average direct effect of the therapeutic alliance on job engagement. Cooperation, comprehension, trust, respect, and shared objectives and ideals are traits of mental health professionals who can form and develop alliances with their clients ([5]). Similarly, clients will be more dedicated to attending sessions if they comply with homework, instructions, obligations, and chores, boosting collaboration and problem-solving ability ([5], [6]). Mental health services providers are undoubtedly better equipped to handle the demands of work, excitement, dedication, and a desire to perform well, as well as to be fully engaged in their work and experience happiness and well-being when their clients are cooperative, responsive, and compliant ([11]; [60]; [8]).

The ability of mental health service providers to develop therapeutic alliances with their clients promotes positive self-disclosure, which is common in therapeutic relationships based on familiarity, affection, trust, and cooperation, which support the mental health service providers in their work and make it easier to gather crucial information from clients about their problems or goals. This allows mental health service providers to diagnose based on evidence and make accurate therapeutic decisions ([14]; [58]). AI systems assist mental health service providers in carrying out these tasks with their clients ([5], [6]).

Based on the present results, we concluded that the predictive potential of the structural model—which includes the use of AI, therapeutic alliance, and job engagement variables—was superior to that of the indicator average model and the linear model of the variables under investigation. The use of AI and generative intelligence techniques also assist mental health service providers in developing a strong therapeutic alliance with their clients, which improves their job engagement, sense of belonging, enthusiasm, dedication, and psychological well-being and decreases work-related burdens, such as time management, diagnosis, and treatment, and increasing the efficacy of the therapeutic intervention system.

### Limitations and Future Directions

The current study used the bootstrap approach to evaluate the mediation role, and the descriptive method to examine the structural model that comprises the use of AI, therapeutic alliance, and job engagement in a sample of mental health service providers. According to the current findings, the predictive power of the structural model is higher than that of the linear model (LM) and the indicator average model (IA). Additionally, the use of AI was found to have a statistically significant overall effect on job engagement and therapeutic alliance. Additionally, it was found that the therapeutic had a median positive statistically significant effect on job engagement.

## 6. Conclusions

The current study used the bootstrap approach to evaluate the mediation role, and the descriptive method to examine the structural model consisted of the use of AI, therapeutic alliance, and job engagement in a sample of mental health service providers. According to the current findings, the predictive power of the structural model is higher than that of the linear model (LM) and the indicator average model (IA). Additionally, the use of AI was found to have a statistically significant overall effect on job engagement and therapeutic alliance. Additionally, it was found that the therapeutic had a median positive statistically significant effect on job engagement.

### Implications

The current study’s findings have many practical aspects and highlight the significance of incorporating interventions to enhance the use of AI applications by mental health service providers in their professional practice. In addition, these results may benefit policymakers and those responsible for managing the mental health services sector in making many decisions regarding:▪Integrating training of mental health service providers, especially new ones, on using AI applications in the job development plan;▪Incorporating AI applications in communicating with beneficiaries of mental health services;▪Using artificial intelligence to analyze large and diverse metadata, formulate a case, suggest a treatment, and develop a therapy plan;▪Integrating AI skills in evaluating the performance of mental health care providers;▪Setting a requirement for proficiency in using AI applications in professional selection.

If these procedures are undertaken, improving the quality of the work environment and reducing the increasing workload on mental health service providers may contribute to reducing burnout, improving the quality of mental health services provided to beneficiaries, and maximizing their benefits.

## Figures and Tables

**Figure 1 behavsci-15-00021-f001:**
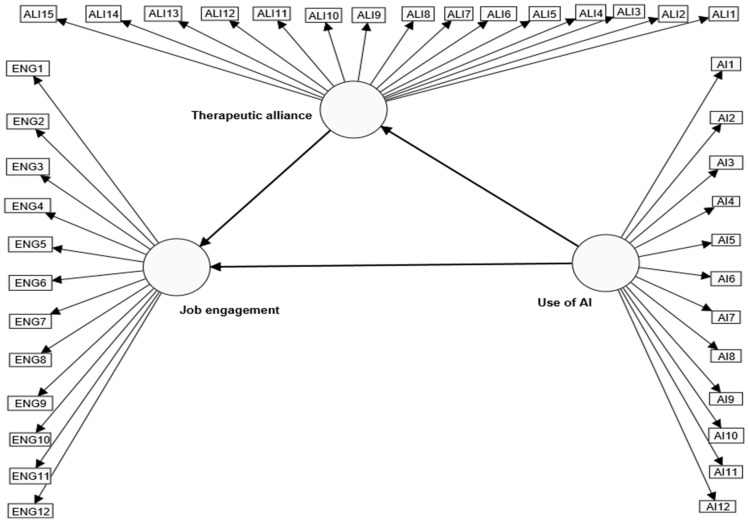
The hypothesized structural model for the use of AI, therapeutic alliance, and job engagement variables.

**Figure 2 behavsci-15-00021-f002:**
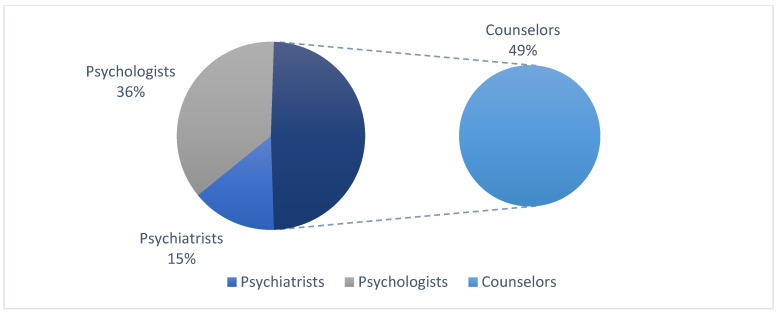
The study sample position distribution.

**Figure 3 behavsci-15-00021-f003:**
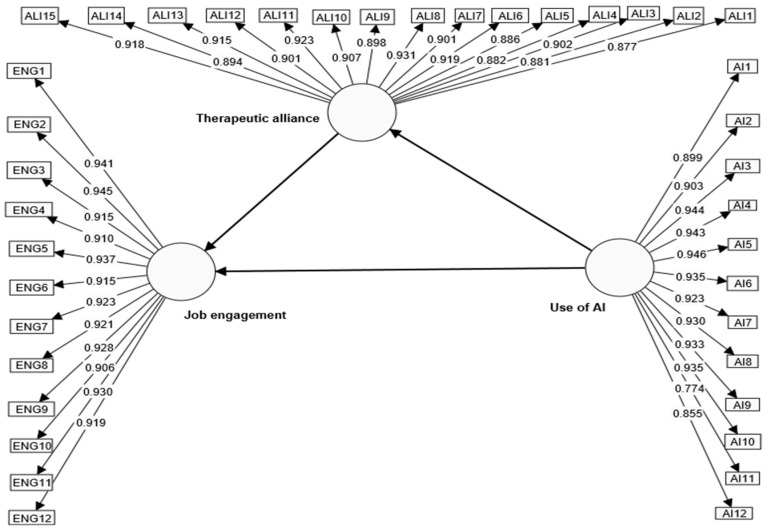
Loading of the study variables’ indicators according to the outputs of the Smart Pls-4 program.

**Table 1 behavsci-15-00021-t001:** Pearson correlation coefficients * between items and the UAIQ as a whole.

Items	1	2	3	4	5	6	7	8	9	10	11	12	UAIQ
1	—												
2	0.770	—											
3	0.857	0.831	—										
4	0.859	0.843	0.891	—									
5	0.857	0.837	0.908	0.920	—								
6	0.850	0.842	0.899	0.878	0.889	—							
7	0.785	0.830	0.860	0.849	0.861	0.861	—						
8	0.830	0.805	0.873	0.857	0.872	0.874	0.868	—					
9	0.822	0.822	0.893	0.865	0.857	0.841	0.853	0.856	—				
10	0.825	0.852	0.895	0.878	0.874	0.851	0.842	0.844	0.910	—			
11	0.654	0.651	0.636	0.694	0.688	0.674	0.691	0.690	0.686	0.666	—		
12	0.708	0.771	0.751	0.761	0.761	0.745	0.775	0.780	0.773	0.759	0.770	—	
UAIQ	0.898	0.903	0.943	0.943	0.945	0.934	0.922	0.929	0.933	0.935	0.777	0.857	—

* All Pearson correlation coefficients are significant at the 0.001 level according to the JASP program output.

**Table 2 behavsci-15-00021-t002:** Pearson correlation coefficients between UAIQ items and their dimension.

Item	Diagnosis and Evaluation	Providing Support	Modeling Practices and Behaviors	Organizing Sessions and Managing Time
1	0.913			
2	0.945			
3	0.938			
4		0.939		
5		0.947		
6		0.940		
7			0.941	
8			0.946	
9			0.920	
10				0.938
11				0.944
12				0.934

**Table 3 behavsci-15-00021-t003:** Pearson correlation coefficients between the dimensions and UAIQ as a whole.

Item	Diagnosis and Evaluation	Providing Support	Modeling Practices and Behaviors	Organizing Sessions and Managing Time	UAIQ
Diagnosis and evaluation	-				
Providing support	0.921	-			
Modeling practices and behaviors	0.928	0.910	-		
Organizing sessions and managing time	0.915	0.937	0.913	-	
UAIQ	0.926	0.932	0.928	0.911	-

**Table 4 behavsci-15-00021-t004:** Pearson correlation coefficients * between items and the TAS as a whole.

Items	1	2	3	4	5	6	7	8	9	10	11	12	13	14	15	TAS
1	—															
2	0.815	—														
3	0.852	0.854	—													
4	0.778	0.830	0.811	—												
5	0.832	0.793	0.860	0.824	—											
6	0.788	0.783	0.781	0.778	0.784	—										
7	0.763	0.769	0.761	0.728	0.733	0.831	—									
8	0.779	0.806	0.819	0.786	0.785	0.854	0.857	—								
9	0.742	0.745	0.771	0.749	0.735	0.837	0.821	0.844	—							
10	0.773	0.758	0.778	0.787	0.792	0.841	0.767	0.852	0.816	—						
11	0.760	0.790	0.787	0.788	0.753	0.858	0.846	0.837	0.857	0.853	—					
12	0.771	0.775	0.799	0.777	0.784	0.812	0.810	0.841	0.775	0.811	0.823	—				
13	0.741	0.745	0.759	0.796	0.799	0.847	0.831	0.845	0.824	0.807	0.864	0.823	—			
14	0.725	0.710	0.779	0.748	0.747	0.815	0.806	0.845	0.809	0.809	0.823	0.789	0.843	—		
15	0.777	0.778	0.821	0.776	0.794	0.822	0.859	0.839	0.808	0.822	0.836	0.803	0.844	0.834	—	
TAS	0.880	0.884	0.905	0.884	0.889	0.918	0.900	0.930	0.896	0.906	0.921	0.899	0.914	0.892	0.918	**—**

* All Pearson correlation coefficients are significant at the 0.001 level according to the JASP program output.

**Table 5 behavsci-15-00021-t005:** Pearson correlation coefficients between items of TAS and their dimension.

Items	Trust and Cooperation	Self-Awareness and Flexible Empathy	Shared Values	Client Compliance	Conflict Management
1	0.912				
2	0.942				
3	0.927				
4		0.933			
5		0.941			
6		0.918			
7			0.899		
8			0.914		
9			0.930		
10				0.924	
11				0.935	
12				0.927	
13					0.913
14					0.939
15					0.905

**Table 6 behavsci-15-00021-t006:** Pearson correlation coefficients between the dimensions and TAS as a whole.

Variable	Trust and Cooperation	Self-Awareness and Flexible Empathy	Shared Values	Client Compliance	Conflict Management	ATS
Trust and collaboration	—					
Self-awareness and flexible empathy	0.894	—				
Shared values	0.903	0.941	—			
Client compliance	0.899	0.934	0.931	—		
Conflict management	0.930	0.917	0.919	0.914	—	
TAI	0.923	0.916	0.929	0.913	0.934	—

**Table 7 behavsci-15-00021-t007:** Pearson correlation coefficients * between items and the JES as a whole.

Items	1	2	3	4	5	6	7	8	9	10	11	12	JES
1	—												
2	0.886	—											
3	0.847	0.874	—										
4	0.827	0.839	0.805	—									
5	0.871	0.859	0.849	0.851	—								
6	0.863	0.854	0.788	0.795	0.845	—							
7	0.881	0.887	0.820	0.803	0.830	0.830	—						
8	0.836	0.850	0.872	0.813	0.848	0.827	0.831	—					
9	0.853	0.879	0.846	0.853	0.845	0.828	0.831	0.849	—				
10	0.836	0.846	0.805	0.841	0.857	0.842	0.782	0.809	0.836	—			
11	0.861	0.851	0.830	0.841	0.875	0.847	0.865	0.820	0.844	0.833	—		
12	0.867	0.852	0.814	0.822	0.857	0.824	0.868	0.864	0.823	0.761	0.836	—	
JES	0.940	0.945	0.916	0.909	0.937	0.915	0.922	0.922	0.928	0.906	0.930	0.921	—

* All Pearson correlation coefficients are significant at the 0.001 level according to the JASP program output.

**Table 8 behavsci-15-00021-t008:** Pearson correlation coefficients between items of JES and their dimension.

Item	Energy and Vitality	Enthusiasm and Dedication	Absorption and Persistence	Well-Being and Belonging to the Job
1	0.943			
2	0.926			
3	0.918			
4		0.911		
5		0.935		
6		0.918		
7			0.925	
8			0.916	
9			0.935	
10				0.930
11				0.915
12				0.905

**Table 9 behavsci-15-00021-t009:** Pearson correlation coefficients between the dimensions and JES total score.

Variable	Energy and Vitality	Enthusiasm and Dedication	Absorption and Persistence	Well-Being and Belonging to the Job	JES
Energy and vitality	—				
Enthusiasm and dedication	0.927	—			
Absorption and persistence	0.916	0.910	—		
Well-being and belonging to the job	0.936	0.926	0.913	—	
JES	0.931	0.928	0.917	0.939	0.928

**Table 10 behavsci-15-00021-t010:** Results of the heterotrait-monotrait ratio (HTMT).

Variables	HTMT
Use of AI <-> Job engagement	0.526
Use of AI <-> Therapeutic alliance	0.622
Therapeutic alliance <-> Job engagement	0.695

**Table 11 behavsci-15-00021-t011:** Variance inflation factors (VIF) for the outer model.

Items	VIF	Items	VIF	Items	VIF	Items	VIF
1	1.599	11	1.504	21	2.568	31	3.319
2	1.539	12	1.602	22	2.349	32	3.395
3	2.542	13	2.209	23	2.545	33	4.402
4	2.543	14	1.925	24	2.431	34	2.273
5	1.443	15	2.128	25	3.212	35	2.529
6	1.496	16	2.205	26	2.667	36	3.012
7	1.388	17	2.159	27	1.831	37	2.888
8	1.788	18	2.168	28	2.727	38	2.568
9	1.426	19	2.269	29	4.406	39	2.6
10	1.307	20	3.436	30	3.622		

**Table 12 behavsci-15-00021-t012:** Variance inflation coefficient VIF for the inner model.

Model	VIF
Use of AI -> Job engagement	1.382
Use of AI -> Therapeutic alliance	1
Therapeutic alliance -> Job engagement	1.382

**Table 13 behavsci-15-00021-t013:** Direct, indirect, and total effects.

Effect	Paths	Effect	SD	*t*-Test	*p*
Direct	Use of AI -> Job engagement	0.277	0.112	2.464	0.014
	Use of AI -> Therapeutic alliance	0.941	0.015	63.767	0.000
	Therapeutic alliance -> Job engagement	0.694	0.113	6.126	0.000
Indirect	Use of AI -> Therapeutic alliance -> Job engagement	0.653	0.103	6.317	0.000
Total	Use of AI -> Job engagement	0.930	0.017	53.404	0.000
	Use of AI -> Therapeutic alliance	0.941	0.015	63.767	0.000
	Therapeutic alliance -> Job engagement	0.694	0.113	6.126	0.000

**Table 14 behavsci-15-00021-t014:** Summary of the CVPAT test for the PLS-SEM model vs IA.

Variable	PLS Loss	IA Loss	Average Loss Difference	T Value	*p* Value
Therapeutic alliance	0.647	1.663	−1.016	12.907	0.000
Job engagement	1.425	2.462	−1.038	13.835	0.000
Total	0.993	2.018	−1.025	16.015	0.000

**Table 15 behavsci-15-00021-t015:** Summary of the CVPAT test for the PLS-SEM model vs. the linear model (LM).

Variable	PLS Loss	LM loss	Average Loss Difference	T Value	*p* Value
Therapeutic alliance	0.647	0.700	−0.053	3.269	0.001
Job engagement	1.425	1.580	−0.156	3.477	0.001
Total	0.993	1.091	−0.099	4.444	0.000

## Data Availability

The original contributions presented in this study are included in the article.

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
