# Peer review of "Causal Relationships Between the Use of AI, Therapeutic Alliance, and Job Engagement Among Psychological Service Practitioners"

_behavsci, 2024, doi:10.3390/bs15010021_

Round 1
Reviewer 1 Report
Comments and Suggestions for Authors
Dear Author(s),
I want to express my appreciation for the well-written and effectively presented paper above, which provides comprehensive data analysis. The article successfully covers a wide range of aspects. However, I suggest that the literature review and hypothesis development are not presented in a structured way. Also, there are gaps while building the argument in the literature review and hypothesis development at a few places. The methodology section has not handled all the biases. It is worth mentioning that the discussion and theoretical implications are well-written, but they can be enriched by including a separate section on managerial implications. Despite these concerns, the paper is of good quality overall. Language is the part that requires much more effort. Nevertheless, this paper merits acceptance with major revision.
I am suggesting some of the points which may help you in revising the paper:
1. Present the literature review and hypothesis development in a more structured way.
2. Include a section on Non-Response Bias and Common Method Bias using a common latent factor.
3. Include a separate section for managerial implications after theoretical implications.
4. Language could be better in several places; it requires professional proofreading.
5. The contribution of the study is not clear and not applicable. Improve the introduction to be clearer and more innovative
6. Please highlight the sampling technique and data collection process.
Comments on the Quality of English Languagemust be improved
Author Response
Dear Author(s),
I want to express my appreciation for the well-written and effectively presented paper above, which provides comprehensive data analysis. The article successfully covers a wide range of aspects. However, I suggest that the literature review and hypothesis development are not presented in a structured way. Also, there are gaps while building the argument in the literature review and hypothesis development at a few places. The methodology section has not handled all the biases. It is worth mentioning that the discussion and theoretical implications are well-written, but they can be enriched by including a separate section on managerial implications. Despite these concerns, the paper is of good quality overall. Language is the part that requires much more effort. Nevertheless, this paper merits acceptance with major revision.
I am suggesting some of the points which may help you in revising the paper:
1. Present the literature review and hypothesis development in a more structured way.
Reply:
Thank you for your comment. We edited this; in the introduction and theoretical background section, we presented many of the results of previous studies on artificial intelligence and its role in providing mental health services (lines 76-83, 92-96, 97-106,120-128) and the challenges facing workers in providing psychological services in developing the therapeutic alliance, as well as presenting the results of studies on the role of artificial intelligence in employee engagement (lines 157-196, 212-224). In the study hypothesis section (lines 225-262), we also explained the gap in previous studies on the object of the current study and derived the theoretical model of the relationships between the use of artificial intelligence, the therapeutic alliance, and the functional integration among psychological service providers (line
Include a section on Non-Response Bias and Common Method Bias using a common latent factor.
Reply:
Thank you for your comment. We have already included the results of Convergent Validity. The results showed that the average variance extracted AVE for using AI, therapeutic alliance, and job engagement was higher than (0.50). Thus, these values are acceptable ​​, according to Hair et al. (2019) (lines 476-479).
Also, we computed the Heterotrait-Monotrait ratio (HTMT) to assess discriminant validity; the findings found that the HTMT values fell within the acceptable standard range per the Henseler et al. (2021) criterion. These results indicated that using AI, therapeutic alliance, and job engagement, latent variables are free from interference, as the relationship between them did not exceed (0.80), which indicated their discriminant validity (lines 480-491).
Additionally, from lines (496 to 508) we wrote the results of the Collinearity by performing the Variance Inflation Coefficient (VIF); the results shown in Tables 11 and 12 indicated that the model could be considered free of common method bias because all of the VIF did not exceed 5, according to Hair et al. (2019).
The results of Convergent Validity and Collinearity indicated that the proposed model in our study is free of response bias and common method bias.
Include a separate section for managerial implications after theoretical implications.
Reply:
Thank you for this suggestion; we added this section as a separate section (lines 690-712).
Language could be better in several places; it requires professional proofreading.
Reply:
Thank you for your comment; we edited the professional proofreading.
- The contribution of the study is not clear and not applicable. Improve the introduction to be clearer and more innovative.
Reply:
Thank you for your comment. We have edited it.
- Please highlight the sampling technique and data collection process.
Reply:
Thank you for your comment. We have edited it.

Reviewer 2 Report
Comments and Suggestions for Authors
The paper is very well structured. The majority of the references are from the last 5 years. It is written in an accessible English language. I believe the subject will be of interest to the research community. The study is replicable in other contexts, as the authors point out. I believe that some aspects of writing should be corrected, as indicated below:
Line 75 - the full stop at the end of the sentence is missing.
Line 82 - a word seems to be missed in the sentence "In terms of treatment, AI applications can play an important by providing insights into the client-therapist interaction and the client’s psychological states [25].". I suggest the term "(...) play an important role (...)".
Line 341 and 343 - The item number doesn't seem correct. Instead of being "(3) Absorption and persistence measure..." it should be "(4) Absorption and persistence measure...". Instead of being "(4) Well-..." it should be "(5) Well-...".
Line 375 - The table number is missing. Instead of "... are displayed in the table below..." it should be " are displayed in the table 2 below..."
Line 381—The table number is missing. It must be mentioned as follows: "... in table 3:...".
Line 403—The table number is missing. It must be mentioned as follows: "... in the table 5 below:...".
Line 410 - The table number is missing. It should be number 6.
Line 440 - Instead of "...in the following table:" it should be "in the following table 9:".
Line 471 - Remove the parenthesis from the number that references the figure.
Line 485 - Add the number to the table as "table 10".
Line 504 - Add the number to the table as "table 11".
Line 507 - Add the number to the table as "table 12".
Line 514 - Instead of "The following table shows the results." it must be "Table 13 shows the results."
Lines 383, 394, 399/400, 406, 414, 424, 430, 436, 443, 487, 510, 517, 530, 570, and 577 - Remove the parenthesis from the number that references the tables.
Line 568 - Instead of "...results are shown in the following two tables:" it must be "...results are shown in the following Tables 14 and 15:"
Author Response
Reply:
Dear Reviewer
Greetings
Thank you for your great efforts in reviewing our paper, we would like to inform you that all of your comments were edited and highlighted with yellow in the revised manuscript.
Best regards

Round 2
Reviewer 1 Report
Comments and Suggestions for Authors
Thank you for your efforts
Comments on the Quality of English LanguageEnglish need improvment
Author Response
Dear esteemed reviewer,
We are grateful for your outstanding efforts in reviewing our manuscript and for the valuable feedback and comments you provided, which contributed to its improvement. We would like to inform you that we have taken your comments into consideration, reviewed the entire manuscript, and made the necessary enhancements. We hope that the revised manuscript will meet your approval and acceptance.
Sincerely
